# OpenReview forum: "Many LLMs Are More Utilitarian Than One"
_NeurIPS.cc/2025/Conference — NeurIPS 2025 poster_

### Official Review · Reviewer_2YLG · 2025-07-01

**Clarity:** 3
**Significance:** 2
**Originality:** 3
**Rating:** 4
**Confidence:** 4

**Summary:**

This paper investigates whether LLMs become more utilitarian in groups. The authors test six different LLMs under two conditions: 1) Solo: models reason alone; 2) Group: models discussed moral dilemmas in pairs or triads. They conduct the test on established moral and psychological questionnaires. The findings show LLMs exhibit a utilitarian boost, particularly in personal dilemmas involving direct harm. The authors also compare these results to human behavior. Though the outcome from LLM groups resembles that of humans, the underlying mechanisms differ. The utilitarian boost in human groups stems from a heightened sensitivity to the consequences of their actions. In contrast, LLM groups show varied mechanisms, such as either reduced sensitivity to moral norms or an enhanced sense of impartiality.

**Questions:**

See weakness.

**Ethical Concerns:**

["NO or VERY MINOR ethics concerns only"]

**Final Justification:**

While I appreciate the authors' acceptance of my suggestions and plans for follow-up experiments, the lack of solid results means I must maintain my original score.

**Limitations:**

yes

**Quality:**

3

**Strengths And Weaknesses:**

# Strengths:

1. This paper focuses on the moral domain in multi-agent systems, which is a significant but often-overlooked research area.
2. The experiment, which takes inspiration from moral psychology (e.g., the OUS and CNI models), seems to be relevant to the focus of the motivation.
3. The study provides an analysis of the mechanisms behind the utilitarian boost. It reveals that different LLMs have distinct "moral change profiles", such as GPT-4.1's alarming "Norm-Violating" tendency. This serves as a crucial warning for deploying multi-agent systems in high-stakes domains.

# Weaknesses:

1. While using only identical agents simplifies experimental control, it also restricts the study’s realism and generalizability. In contrast to human groups, which are typically characterized by diversity in perspectives, reasoning styles, and decision-making heuristics, a multi-agent system composed of identical LLMs may fail to capture the complex dynamics arising from heterogeneous interactions.
2. This paper brings limited insights to the community. The current version only conducts experiments based on human-oriented protocols and reports the results of some LLMs. More experiments and discussions are needed to, for example, explain the inter-model differences, guide the design, learning, or inference of LLMs.
3. It is intriguing that the increase in GPT-4.1’s utilitarian boost is only measured in triads. However, the increase in GPT-4.1's moral acceptability score is not significant, and the score remains at a consistently low level. The current investigation into GPT-4.1 is not sufficiently thorough. The study should be supplemented with results from GPT-4.1 in a group of size 4.

# Minor issues:

1. Typos: In the "Moral Change Profile" radar plot, "aving-Desant" is likely a typo for "Saving-Deontological", and "actual-Other-Uslitarian" is likely a typo for "Factual-Other-Utilitarian.”
2. The format of references should be improved.

---

> ### Author Rebuttal · Authors · 2025-07-31
>
> We appreciate the reviewer’s recognition of the significance of multi-agent moral reasoning, the mechanistic explanation, and our efforts to contribute to LLM collective reasoning. Your positive evaluation of the experimental design and framing is deeply encouraging.
>
> **[W1]** While using only identical agents simplifies experimental control, it also restricts the study’s realism and generalizability. In contrast to human groups, which are typically characterized by diversity in perspectives, reasoning styles, and decision-making heuristics, a multi-agent system composed of identical LLMs may fail to capture the complex dynamics arising from heterogeneous interactions.
>
> **[R1]** While human experiments inspired our study and the tools used to explore the Utilitarian Boost's mechanisms, simulating real-world human behavior was not our focus. We were instead focused on alignment issues within multi-agent LLM systems designed to tackle moral reasoning. However, we agree with the reviewer that diversity in agent perspectives provides a richer understanding of how a Utilitarian Boost emerges and may be adjusted. The novel experiments described in response to the following comment were designed to elucidate such effects in the mind.
>
>
> **[W2]** This paper brings limited insights to the community. The current version only conducts experiments based on human-oriented protocols and reports the results of some LLMs. More experiments and discussions are needed to, for example, explain the inter-model differences, guide the design, learning, or inference of LLMs.
>
> (Note for the (Senior) Area Chairs, the answer R2 below is similar to the answer we gave to reviewer iiw5 under R1, but there is one additional paragraph in the middle).
>
> **[R2]** To provide guideposts for the design of moral multi-agent LLMs, we performed a series of post-hoc experiments and analyses that explored how different model choices, prompts, group setups and discussion formats affect the Utilitarian Boost. We discuss the results in more detail below.
>
> We systematically varied group composition across three dimensions:
> - Model size to account for sophistication: small, medium and large checkpoints for the same architectures (e.g., Qwen2.5-instruct:7B, 32B, 72B)
> - Model family to account for training and design differences: Pairing models with distinct baseline utilitarianism scores and effect-sizes, demonstrating distinct output behaviors
> - Moral framework to account for alignment differences: Through role prompts that instilled utilitarian, deontological or neutral approaches.
>
> We will report the detailed results of all these experiments for dyads and triads in the revised manuscript. The overall takeaway is that diversity in group composition, be it introduced via prompts, architecture or sophistication attenuates the Utilitarian Boost.
> This offers model developers several avenues to enhance or reduce the Boost as needed, as we shall explain. Critically, we believe that the profiles calculated for each LLM (e.g. Figures 3, 5 and 6), are helpful for alignment teams with detailed knowledge of their pretraining and posttraining recipes. (For example, the Qwen2.5, Qwen3, and QwQ models all have similar architecture and size but exhibit different profiles which will come down to the recipes).
>
> Here, we would like to emphasize two other dimensions that are just as promising for explaining the Utilitarian Boost and providing ways to manipulate it: The CNI framework and emotional argumentation.
>
> The CNI model (Gawronski et al., 2017) treats utilitarian choice as the joint product of three latent weights: Consequence sensitivity (C), Norm sensitivity (N), and Inaction bias (I). In humans, Rokosz et al. (2025) found that group discussion raises only C, leaving N and I unchanged. The CNI framework shows that human group discussion boosts utilitarianism solely by increasing consequence‑sensitivity (C), with no change in norm‑sensitivity (N) or omission bias (I) (ΔG² = 55.6, p < .001, d ≈ 1.0). In contrast, LLMs display model‑specific trade‑offs between consequence focus and rule respect as described in the manuscript. Encouraging or discouraging this focus via role prompts could help strengthen or attenuate the Utilitarian Boost as needed.
>
> We also applied post‑hoc probes to the arguments (which we will add to the revised manuscript). Psychology and neuroscience have identified emotions as key to the processing of personal versus non-personal moral dilemmas (Greene et al, 2014). We therefore tagged each argument using a RoBERTa-based (Hartmann, 2022) classifier for Ekman’s six emotions plus neutral. We then correlated those tags with the models’ utilitarian shifts from the Solo to the Group condition. Examining the dominant emotion by experimental condition and dilemma effect revealed an interesting pattern: in the Solo condition, the top emotion label was ‘disgust’, particularly in personal dilemmas where solo judgments were less utilitarian. In contrast, in the Group condition, dilemmas with no or negative utilitarian shift were predominantly labeled ‘neutral’, while those with firm utilitarian shifts were labeled ‘fear’. Tabulating the emotions by LLM, Gemma3 emerges as having the most ‘fear’ label, while showing the most substantial utilitarian boost (Table 1). This finding aligns with human data: both fear and disgust significantly predict utilitarian biases in personal moral dilemmas in humans. For instance, individuals with poorer disgust decoding are more likely to make utilitarian choices in Personal dilemmas (Martina Carmona-Perera et al, 2014). Stress significantly increased utilitarian decisions when participants faced personal moral dilemmas (Farid Yousef, 2012).
>
> These results highlight emotion as a plausible lever for influencing LLM moral reasoning toward utilitarian outcomes, while proposing a potential mechanism at play. Experimentally attenuating disgust or amplifying fear within model outputs, or human persuasive texts, could plausibly shift moral reasoning along the utilitarian-deontic spectrum. This offers a target for future interventions. Taken together, both the CNI model and the use of emotions in the arguments emerge as promising explanatory factors that control the utilitarian boost across different conditions and LLMs.
>
>
> **[W3]** It is intriguing that the increase in GPT-4.1’s utilitarian boost is only measured in triads. However, the increase in GPT-4.1's moral acceptability score is not significant, and the score remains at a consistently low level. The current investigation into GPT-4.1 is not sufficiently thorough. The study should be supplemented with results from GPT-4.1 in a group of size 4.
>
> **[R3]** We thank the reviewer for this suggestion. We are currently performing a follow-up experiment with GPT-4.1 in a group of size 4.

---

> > ### Comment · Reviewer_2YLG · 2025-08-05
> > **Thanks for the response**
> >
> > While I appreciate the authors' acceptance of my suggestions and plans for follow-up experiments, the lack of solid results means I must maintain my original score.

---

> > > ### Author Response · Authors · 2025-08-06
> > > **Follow-up experimnets**
> > >
> > > We thank the reviewer for suggesting follow-up experiments. We tested a group size of 4 (tetrads) in GPT-4.1. Using the same statistical model, and consistent with the triad results, tetrads showed a utilitarian boost relative to solo (b = 0.55, SE = 0.24, z = 2.24, p = .025; 95% CI [1.07, 2.78]). The evidence indicates that the Utilitarian Boost is highly reliable, appearing across all models we tested and replicating at group sizes 3 and 4 for GPT-4.1.
> > > To explore diverse groups, in another experiment, we designed prompts that instructed models to adopt a deontological, utilitarian, or neutral moral orientation, then placed them in homogeneous (DD) or mixed pairs (UD, DU) to test heterogeneous groups. Since we were interested in a Utilitarian Boost and the UU condition could not show such a result due to expected ceiling effects, we excluded utilitarian‑utilitarian (UU) pairs. We compared the first round of discussion with the group reflection phase response. Pairs instructed to reason deontologically (DD) gave higher ratings after discussions of direct harm, demonstrating the robustness of the Utilitarian Boost (Pair - Round1 = +0.377, p = .010). Using step-based analysis, we found group composition as moderators: mixed UD/DU pairs became less utilitarian in the last round compared to the first (Last-First Round = −0.323, p < .0001), indicating that initial moral framework diversity counters the Utilitarian Boost during the discussion phase.  Additional analyses showed a very interesting dynamic: utilitarian agents are more prone to ‘change of mind’ than deontological ones (D-U = -0.467, p < .0001). This mirrors human data in moral conformity: individuals with utilitarian leanings tend to conform both more often and more strongly than their deontological counterparts (Marton-Alper et al. 2022). People also tend to conform more readily to deontological than to utilitarian opinions, revealing an asymmetry in moral conformity (Bostyn & Roets, 2017), which mimics our post-hoc experimental results in our diverse settings.
> > > In another experiment, we mixed the model family to account for training and design differences by pairing models with distinct baseline utilitarianism scores and effect sizes. We directly compared the heterogeneous (mixed-group) condition with its solo baseline for the same LLMs using the same cumulative link mixed model as in the submitted manuscript. The mixed-group condition showed higher odds of utilitarian endorsements than solo, b = 0.154, SE = 0.084, z = 1.84, directional one-tailed p = .03 (a priori), 95% CI [0.99, 1.37]. When directly contrasting group types, the most substantial boost was observed in homogeneous (same-model) groups, which were significantly more utilitarian than heterogeneous groups (same > mixed: b = 0.305, p < 0.001, 95% CI [1.15, 1.51]). This indicates that the trajectory of diverse groups from Step 1 to Step 7 may differ from that of homogeneous groups, thereby mitigating the effect, but does not undermine the group-versus-solo effect in general. Accordingly, both heterogeneous and homogeneous groups were more utilitarian than the solo baseline, underscoring the robustness of the group effect.
> > > Overall, the evidence indicates that the Utilitarian Boost is robust in multi-agent groups, appearing across all models we tested. It can be mitigated under specific, identifiable conditions and otherwise emerges reliably. We therefore view these boundary conditions (e.g., model diversity in groups, initial prompts/role instructions) as practical levers to predict, modulate, and, when needed, prevent the Utilitarian Boost in future multi-agent systems.

---

### Official Review · Reviewer_32Uz · 2025-07-02

**Clarity:** 3
**Significance:** 3
**Originality:** 3
**Rating:** 4
**Confidence:** 3

**Summary:**

This paper studies collective moral decision-making using large language models (LLMs). Specifically, the authors investigate a known utilitarian boost from human studies and whether this effect also occurs in multi-agent systems with the ability to collaborate over multiple turns of conversation. The results span six different models tested on different moral dilemmas sourced from various datasets. Interestingly, while the authors also find a utilitarian boost in LLM multi-agent systems, they suggest that the underlying mechanisms are different than in human groups based on profiles generated from survey-like questionnaires. This work is relatively important for providing a framework for studying alignment of multi-agent systems as well as computational ethics.

**Questions:**

I have the following questions:

1. The authors only looked at homogeneous groups of agents in a collaborative setting based on a single model at a time. Do they have thoughts how the results might change with heterogenous groups of agents that might be aligned slightly differently, reflecting the diversity of values and opinions of the general human population (e.g. modeling different user personas)?

2. If would be useful to think of potential ablation studies, e.g. prompting a model to act specifically in a non-utilitarian way and measuring the impact on the moral acceptability rating or re-using prior responses from previous turns. These results could reveal how important particular responses are in a pair/triad and the impact they have on the final ratings.

3. How was the number of rounds (6) of inter-agent exchange chosen and how sensitive are the results to this? The results suggest a utilitarian boost after many rounds, but how does this evolve across the different rounds?

**Ethical Concerns:**

["NO or VERY MINOR ethics concerns only"]

**Final Justification:**

The authors performed additional experiments that help clarify under which conditions the utilitarian boost appears; however, I still have some concerns about the robustness of the effect.

**Limitations:**

The paper includes a dedicated limitations section, which addresses several points. One potential dual use concern is how this framework could be used to potentially generate misinformation and influence group thinking (e.g. social engineering). Another potential limitation that should also be addressed is the ability of LLMs to "faithfully" answer survey questions (e.g. questionnaires), as there is a lot of recent work that questions how valid the generated model outputs are, e.g. see:

Dominguez-Olmedo, R., Hardt, M., & Mendler-Dünner, C. (2024). Questioning the survey responses of large language models. Advances in Neural Information Processing Systems, 37, 45850-45878.

**Paper Formatting Concerns:**

No concerns.

**Quality:**

3

**Strengths And Weaknesses:**

**Strengths**: Overall, this paper studies the dynamics of collective moral judgment and the impact of a utilitarian boost, which has been found in human studies. These are interesting and relevant questions, as LLMs are increasingly being used in new contexts, with a potentially large impact in studying whether this phenomenon also exists in LLM-based multi-agent systems. The paper contains quantified experiments across multiple models (including both open access and proprietary models) and datasets, demonstrating generalization of the results. The analyses also are statistically sound and methods/tools are well described in the Appendix of the paper.

**Weaknesses**:  The paper has the following weaknesses:

- It's unclear to me if this observed effect only applies to moral ratings/acceptability, or if it is a reflection of LLMs' tendency to "change their minds" based on collective interaction and debate. Quantifying similar changes in decisions over morally neutral scenarios may help to disentangle this, and adding additional discussion to the paper on this may be useful.

- The authors leverage different survey tools to understand the mechanisms of the utilitarian boost (e.g. OUS, CNI, etc.) However, it is well known that LLMs may not provide meaningful responses to survey questions, as they have certain biases related to position of choices, prompt structuring, etc. It would be good to better note this limitation of using survey tools to evaluate LLMs and how it might impact conclusions.

- As different LLMs show different changes in the emphasis on moral values (Figure 3), it's unclear the source of the utilitarian boost across models as there is no clear trend in profiles. This is somewhat unsatisfying, as it muddies the picture of what exactly causes the utilitarian boost across models. The authors also suggest that these LLM profiles are different from humans, but the authors only cite prior work and do not map human profiles onto similar charts as the LLMs.

- It appears that models self-report their moral acceptability rating after each turn of conversation, and it is known that self-reporting may often have biases (as in humans). Did the authors look into using an external evaluator (human or LLM) to assess an LLM's generated reasoning and choice and the moral acceptability of that?

---

> ### Author Rebuttal · Authors · 2025-07-31
>
> We thank the reviewer for their insights. We are glad they found the paper's contribution strong. We address their further comments below.
>
> [W1] The effect may reflect general mind-changing in LLMs, not specific to moral reasoning
>
> [R1] We tested three sets of scenarios validated by Greene 2014: non-moral scenarios (low-stakes action judgments), contrasted with indirect and direct harm moral scenarios. Our mixed effects model showed no Utilitarian Boost in non‑moral scenarios (dyads: t = 1.92, p = .39; triads: t = 1.92, p = .39). Only personal dilemmas that involve direct harm to maximize utility showed significant changes. This emerged across all LLMs, indicating that the Utilitarian Boost is tied to collective interaction over moral acceptability rather than general LLM judgment variability in the groups. Equivalence testing (TOST) confirmed the non-moral difference was negligible (p=.002). We highlighted this distinction more explicitly in the revised manuscript.
>
> [W2] highlight the limitations of using survey tools to evaluate LLMs
>
> [R2] We have added the nuances of the LLM survey response in the limitations section.
>
> [W3.a] Where does the utilitarian boost come from? No clear trend in profiles.
>
> [R3.a] The Utilitarian Boost is reflected in the Solo vs Group moral acceptability differences within Figure 2. Despite some differences in effect size, all models show the same trend both in dyads and triads. But a variety of mechanisms could produce the same effect, which are the focus of the profiles in Figure 3. The diversity of the profiles suggests that different models reach the same effect through different reasoning pathways. The most discriminant distinctions come from the CNI model (Gawronsky et al., 2017), which shows different levels of Consequence sensitivity, Norm sensitivity, and Inaction bias across models. This provides a process-level explanation for the different profiles. However, this is not to say that LLMs are exhausting all the possible pathways for generating a Utilitarian Boost identified in psychological studies. Therefore, we performed several post-hoc experiments and analyses, which will be added to the Appendix in the revised version.
>
> Building on evidence that emotions shape moral judgment (Greene et al, 2014). We used a RoBERTa-based (Hartmann, 2022) classifier to tag arguments with Ekman’s six emotions + neutral, and correlated tags with utilitarian shifts from Solo to Group condition. A clear pattern emerged: in Solo, ‘disgust’ was the top emotion where judgments were less utilitarian. In contrast, Group condition responses with strong utilitarian shifts were labeled ‘fear’. These findings align with human data: both fear and disgust significantly predict utilitarian biases in moral dilemmas in humans. For instance, individuals with poorer disgust decoding are more likely to make utilitarian choices in personal dilemmas (Perera et al., 2014).
>
> [W3.b] Mapping human profiles onto similar charts as the LLMs.
>
> (Note for the (Senior) Area Chairs, the answer R3b below is also given identically to reviewer CAnQ under R1.)
>
> [R3.b] Our study is focused on alignment issues within multi-agent LLM systems rather than the simulation of human group behavior in similar tasks. We hence believe that the causes and moderators of the utilitarian boost in multi-agent LLMs remain scientifically and practically important regardless of (dis)similarity to human groups. However, we agree with the reviewer that comparisons with humans are essential for understanding the effect. We will add additional information about prior human experiments to the revised manuscript in a study-by-study manner to allow for more fine-grained comparisons. We have summarized these comparisons below.
>
> (i) Keshmirian et al. (2021): individual ratings cluster a little below the scale midpoint (≈ 3.8 / 7). Across the six LLMs, Solo means sit lower (≈ 2.7 – 3.6), so agents start less utilitarian than people. This baseline gap is most significant for GPT‑4.1 (2.66) and smallest for QwQ (3.56). Human groups and every LLM model show the same qualitative pattern: collective discussion pushes judgments toward the utilitarian pole. Quantitatively, Gemma3 produces a larger jump (+0.85) than the human average. The mean boost across models (≈ +0.42) is very close to the human +0.5‑point uptick.
>
> (ii) Cecru et al. (2019): Individual ratings’ mean is 4.92 utilitarian endorsements. Deliberation lifted that to  5.52 (+0.6 points). LLMs show the same qualitative pattern: models’ mean rating increases when in dyads or triads, but the baseline is lower (2.7‑3.6 on the same scale).
>
> (iii)  Rokosz et al (2025): group discussion boosts utilitarianism by increasing only C (outcome) in CNI. Norm focus and omission bias stay stable. Pooling all six models gave a β = 2.12 for the Group > Solo contrast. Humans show the same direction but a smaller magnitude: in Rokosz et al.’s triads, the mean “norm‑violating, outcome‑maximising” score rose from 2.48 ± 1.31 to 3.48 ± 1.31 (0–6 scale). LLM groups amplify utilitarianism equally for lethal and resource dilemmas, though individual models diverge.
>
> [W4] Using an external evaluator given the risk of bias in model responses.
>
> [R4] To account for this, we ran a validation study, asking multiple human raters to assign to each argument a moral acceptability rating based on its content. The raters' responses strongly agreed with the models' provided ratings, showing reasonable LLM argumentation and reliable use of the scale. Please see in the current manuscript the Appendix section “Reliability Check: Crowdsourcing”.
>
> [Q1]  How would using heterogeneously (e.g., diverse agent personas) change the results?
>
> [RQ1] We agree with the value of diverse group setups for understanding the dynamics that give rise to or limit the Utilitarian Boost. We therefore performed an additional experiment based on the reviewer's insight that yielded fascinating results.
>
> We systematically varied group composition across three dimensions:
> - Model size to account for sophistication: small, medium, and large checkpoints for the same family of LLMs (e.g., Qwen2.5-ins:7B, 32B, 72B)
> - Model family to account for training and design differences: pairing models with distinct baseline utilitarianism scores and effect sizes.
> - Moral orientation to account for alignment differences: through role prompts that instilled utilitarian, deontological, or neutral approaches.
>
> We will report the detailed results of all these experiments for dyads and triads in the revised manuscript. The overall takeaway is that diversity in group composition, be it introduced via prompts, architecture, or sophistication, attenuates the Utilitarian Boost. This offers model developers several avenues to enhance or reduce the Boost as needed. Details of the third manipulation listed above are included in response to the following comment.
>
> [Q2] Ablate: prompt non-utilitarian or reuse prior answers.
>
> [RQ2] Given time and space limitations, we note fine-grained ablations as an important area for future research in the revised discussion. Prompt experiment results are added as discussed below.
>
> We designed prompts that instructed models to adopt a deontological, utilitarian, or neutral moral orientation, then placed them in homogeneous (DD) or mixed pairs (UD, DU). Since we were interested in a Utilitarian Boost and the UU condition could not demonstrate such a result due to expected ceiling effects, we excluded utilitarian‑utilitarian (UU) pairs. We compared the first round of discussion with the group reflection phase response.
>
> Pairs instructed to reason deontologically (DD) gave higher ratings after discussions of direct harm, demonstrating the robustness of the Utilitarian Boost (Pair - Round1 = +0.377, p = .010).
>
> We found group composition as moderators: mixed UD/DU pairs became less utilitarian (Pair-Round1 = −0.323, p < .0001), indicating that initial moral framework diversity counters the Utilitarian Boost.  Additional analyses showed that utilitarian agents are more prone to conformity than deontological ones (D-U = -0.467, p < .0001). This, interestingly, mirrors human data in moral conformity: individuals with utilitarian leanings tend to conform both more often and more strongly than their deontological counterparts (Marton-Alper et al. 2022). People tend to conform more readily to deontological than to utilitarian opinions, revealing an asymmetry in moral conformity (Bostyn & Roets, 2017).
> Thus, collective moral outcomes of LLM groups are sensitive to the distribution of initial value positions, and a single strategically placed dissenting voice can dampen runaway utilitarianism. This provides practical levers for value‑alignment interventions.
>
> [Q3] Are the results sensitive to the round numbers?
>
>
> [RQ3] We chose six rounds to approximate the amount of discussion generated in similar previous human experiments (e.g., Keshmirian et al., 2021). This allowed us to use similar tools for exploring the mechanisms behind the Utilitarian Boost in LLMs. The results of the step-by-step analysis are added to the revised manuscript. Using mixed effect ordinal regression, the only reliable change in utilitarian rating is the cumulative gain from Step 1 to Step 6 (Δ = –0.161, p = .016). Within that span, the early rise (Step 1 → 3: Δ = –.087, p = .57) and the late interval (Step 3 → 6: Δ = –.074, p = .67) are both small. In other words, almost all of the utilitarian boost crystallises by the third exchange and then almost plateaus. This suggests that extending the dialogue beyond three rounds will likely yield no further change. We are currently running a follow-up experiment where models are asked to deliberate after each step, to allow for even more fine-grained analysis of the pattern.

---

> > ### Comment · Reviewer_32Uz · 2025-08-04
> > **Response to Authors**
> >
> > I have read the author's rebuttal, and thank them for their effort in addressing my feedback and questions. Their responses help me with better understanding the proposed approach and corresponding results. I will raise my score to a 4.
> >
> > The additional grounding in previous human experimental results and analyses with different models, prompts, etc. provide a better understanding of under which conditions the utilitarian boost appears. However, I still have some concerns about the robustness of the observed effect, as it seems to be strongest after a set number of turns and in homogeneous groups of LLMs, with different LLMs seeming to show different underlying mechanisms, without a clear explanation of why this is.

---

> > > ### Author Response · Authors · 2025-08-06
> > > **Boundary Conditions**
> > >
> > > We thank the reviewer for the positive feedback on our rebuttal. To further assess robustness in heterogeneous groups, we directly compared the heterogeneous (mixed-group) condition with its solo baseline for the same LLMs using the same cumulative link mixed model as in the submitted manuscript. The mixed-group condition showed higher odds of utilitarian endorsements than solo, b = 0.154, SE = 0.084, z = 1.84, directional one-tailed p = .03 (a priori), 95% CI [0.99, 1.37]. However, when directly contrasting group types, the most substantial boost was observed in homogeneous (same-model) groups, which were significantly more utilitarian than heterogeneous groups (same > mixed: b = 0.305, p < 0.001, 95% CI [1.15, 1.51]). This indicates that the trajectory of diverse groups from Step 1 to Step 7 may differ from that of homogeneous groups, thereby mitigating the effect, but does not undermine the group-versus-solo effect in general. Accordingly, both heterogeneous and homogeneous groups were more utilitarian than the solo baseline, underscoring the robustness of the group effect. We also tested a group size of 4 (tetrads) in GPT-4.1, given that the effect was less pronounced in GPT-4.1 pairs. Using the same statistical model, and consistent with the triad results, tetrads showed a utilitarian boost relative to solo (b = 0.55, SE = 0.24, z = 2.24, p = .025; 95% CI [1.07, 2.78]).
> > > Overall, the evidence indicates that the Utilitarian Boost is reliable in multi-agent groups, appearing across all models we tested. The effect can be mitigated under specific, identifiable conditions and otherwise emerges reliably. We therefore view these boundary conditions (e.g., model diversity in groups, initial prompts/role instructions) as practical levers to predict, modulate, and, when needed, prevent the Utilitarian Boost in future multi-agent systems.

---

> > > > ### Comment · Reviewer_32Uz · 2025-08-06
> > > > **Response to Authors**
> > > >
> > > > Thank you for clarifying and providing additional supporting information, I will maintain my current rating.

---

### Official Review · Reviewer_iiw5 · 2025-07-02

**Clarity:** 3
**Significance:** 2
**Originality:** 3
**Rating:** 3
**Confidence:** 4

**Summary:**

This paper studies moral judgements of LLMs. They test different models on a set of moral dilemmas, in two settings: solo, where each LLM is reasoning on its own, and group, when there is a group of LLMs and they discuss their choices together. They show that LLMs decide for to maximize the over-all well-being (even if it means to harm one another person) when they reason as a group.

**Questions:**

- It would be interesting to see joint human-LLM or human-in-the-loop experiments.
- In the experiments, does a group of LLMs consist of only one model? How does it differ when we have stronger vs weaker models in a group? Maybe stronger models are more persuasive when they are grouped with weaker ones.

- In a group scenario, we have multi-step debate and discussion, and models have a chance to refine their answers. If we have a multi-step self-debate for a solo model, would its answer change?

**Ethical Concerns:**

["NO or VERY MINOR ethics concerns only"]

**Final Justification:**

I find the observations in this paper interesting, particularly if the authors follow through on their plan (as noted in the rebuttal) to include additional models and groups. However, I am not fully convinced that the contributions, in their current form, are substantial enough for a NeurIPS technical paper.

**Limitations:**

Yes.

**Quality:**

3

**Strengths And Weaknesses:**

- They target a very important problem in AI safety, how LLMs reason as a group in moral dilemmas and how much their reasoning behavior is similar to humans. This is important especially if we rely on majority voting or multi-agent debates for more complex moral situations.

- The paper is well-written and easy to follow.
- They experiment with a variety of moral dilemmas.

- Even though their observation is interesting, it shows limited mechanistic interpretability. Authors tried to investigate why this behavior emerges using psychological scales, but it doesn’t explain why it emerges, and how much it is sensitive to model selection, prompt alternation, etc.

- The scenarios are abstract and modeled after psychology experiments. While suitable for controlled study, they lack connection to concrete real-world applications where such group decisions could play out.

---

> ### Author Rebuttal · Authors · 2025-07-31
>
> We are glad that the reviewer found our topic and the breadth of dilemmas explored as strong features. We address their other comments below.
>
> [W1]: Interesting observation, but offers little mechanistic insight and sensitivity to model or prompt variations.
> [R1]: To understand the mechanisms and sensitivity of the effect to model or prompt variations, we performed a series of post-hoc experiments and analyses. We systematically varied group composition across three dimensions:
>
> - Model size to account for sophistication: small, medium, and large checkpoints for the same architectures (e.g., Qwen2.5-instruct:7B, 32B, 72B)
> - Model family to account for training and design differences: Pairing models with distinct baseline utilitarianism scores and effect sizes, demonstrating distinct output behaviors
> - Moral framework to account for alignment differences: Through role prompts that instilled utilitarian, deontological, or neutral approaches.
>
> We will report the detailed results of all these experiments for dyads and triads in the revised manuscript. The overall takeaway is that diversity in group composition, be it introduced via prompts, architecture, or sophistication, attenuates the Utilitarian Boost. This offers model developers several avenues to enhance or reduce the Boost as needed.
>
> Here, we would also like to emphasize two other dimensions that are just as promising for explaining the Utilitarian Boost and providing ways to manipulate it: The CNI framework and emotional argumentation.
>
> The CNI model treats utilitarian choice as the joint product of three latent weights: Consequence sensitivity (C), Norm sensitivity (N), and Inaction bias (I). In humans, Rokosz et al. (2025) found that group discussion raises only C, leaving N and I unchanged. The CNI framework shows that human group discussion boosts utilitarianism solely by increasing consequence‑sensitivity (C), with no change in norm‑sensitivity (N) or omission bias (I) In contrast, LLMs display model‑specific trade‑offs between consequence focus and rule respect as described in the manuscript. Encouraging or discouraging this focus via role prompts could help strengthen or attenuate the Utilitarian Boost as needed.
>
> We also applied additional post‑hoc probes to the arguments, which will be added to the revised manuscript. Psychology and neuroscience have identified emotions as key to the processing of personal versus non-personal moral dilemmas (Greene et al, 2014). We therefore tagged each argument using a RoBERTa-based (Hartmann, 2022) classifier for Ekman’s six emotions plus neutral. We then correlated those tags with the models’ utilitarian shifts from the Solo to the Group condition. Examining the dominant emotion by experimental condition and dilemma effect revealed an interesting pattern: in the Solo condition, the top emotion label was ‘disgust’, particularly in personal dilemmas where solo judgments were less utilitarian. In contrast, in the Group condition, dilemmas with no or negative utilitarian shift were predominantly labeled ‘neutral’, while those with firm utilitarian shifts were labeled ‘fear’. Tabulating the emotions by LLM, Gemma3 emerges as having the most ‘fear’ label, while showing the most substantial utilitarian boost (Table 1). This finding aligns with human data: both fear and disgust significantly predict utilitarian biases in personal moral dilemmas in humans. For instance, individuals with poorer disgust decoding are more likely to make utilitarian choices in Personal dilemmas (Martina Carmona-Perera et al, 2014). Stress significantly increased utilitarian decisions when participants faced personal moral dilemmas (Farid Yousef, 2012).
>
> These results highlight emotion as a plausible lever for influencing LLM moral reasoning toward utilitarian outcomes, while proposing a potential mechanism at play. Experimentally attenuating disgust or amplifying fear within model outputs, or human persuasive texts, could plausibly shift moral reasoning along the utilitarian-deontic spectrum. This offers a target for future interventions.
>
> Taken together, both the CNI model and the use of emotions in the arguments emerge as promising explanatory factors that control the utilitarian boost across different conditions and LLMs. In contrast, semantic clustering of dilemmas and mapping them to mean effects revealed no consistent patterns. Similarly, mapping argument complexity to utilitarian shifts did not yield meaningful insights.
>
> [W2] The abstract psychology-style scenarios aid control but don’t map neatly onto real-world group decisions.
> [R2] While human experiments provided the inspiration for our study and the tools used to explore the Utilitarian Boost's mechanisms, simulating real-world human behavior was not our focus. We were instead focused on alignment issues within multi-agent LLM systems designed to tackle moral reasoning. On this front, our study balanced two aims:
>
> (i) experimental clarity, retaining well-understood structures that let us pinpoint the processes behind utilitarian shifts to guide interventions on multi-agent LLM systems,
> (ii) ecological validity: grounding moral trade-offs in situations that such models reasonably should be able to handle.
>
> We used two different stimulus sets, one for each of the goals described above:
>
> 1. Classic trolley-type dilemmas give the field’s cleanest test of deontological vs. utilitarian trade-offs, offering a well-mapped baseline for any new agent, with rich psychological data on different mechanisms in individuals and groups.
>
> 2. Körner et al.’s (2023) factual dilemmas translate the same conflict into policy debates based on actual historical events. These scenarios were specifically designed to address real-world group decisions. Redefining death for organ donation, wartime deception, pandemic quarantine, so the task speaks to real committees, ethics boards, and families who must weigh lives against rules are examples of included situations. Körner et al. empirically show that participants rate these scenarios as more realistic, less absurd, and more engaging than classic sacrificial thought experiments, while still cleanly instantiating the deontological-versus-utilitarian conflict and avoiding the common “action = harm” confound.
>
> Together, our results demonstrate how multi-agent LLM deployments can amplify - or dampen - particular moral tendencies in contexts that already mirror real-world collective choices, while also suggesting mechanism-driven ways to control such behavior.
>
> [Q1] It would be interesting to see joint human-LLM or human-in-the-loop experiments.
> [RQ1] We agree that a logical next step to our study is to test mixed teams in which one or more people reason alongside one or more models. Such “human + LLM” panels would let us observe whether the model’s analytic style amplifies or tempers human emotion and how perceived responsibility for the judgment is shared with the LLM. It would also mimic an alternative format in which moral LLMs are commonly used. Because many practical deployments already put users in conversation with LLM as “moral experts,” controlled hybrid‑group experiments can directly inform safer, more transparent applications in future work. We will discuss the promise of this type of design in the revised manuscript.
>
> [Q2] Does a group of LLMs consist of only one model? What about weak vs strong?
> [RQ2] We deployed single-model multi-agent LLM setups to better align our experiments with the standard design of such systems. However, we agree with the reviewer that the testing of heterogeneous groups composed of different models and varying strengths is important for better understanding the Utilitarian Boost. To this end, we ran several post-hoc probes, which we will add to the revised manuscript.
>
> Our heterogeneous model groups were composed of the following sets, chosen for variability in baseline ratings and Utilitarian Boost effect sizes: GPT-4.1 × Qwen3-32B-instruct (Qwen3 variant), Gemma2-27B × Qwen2.5-32B-instruct, and GPT-4.1 × QwQ-32B, Gemma2-27B × QwQ-32. The results show that model heterogeneity dampens the Utilitarian Boost (β = –0.30 ± 0.08, z = –3.79, p = .0001). Nonetheless, homogeneous groups remain more utilitarian than Solo instances of the same model (β = +0.29 ± 0.07, z = 4.24, p = .0001).
>
> As suggested by the reviewer, we also used Strong vs Weak combinations: Gemma 2 27B x Gemma 2 9B, Qwen 2.5 instruct 32B x Qwen 2.5 instruct 72B, Qwen 2.5 instruct 7B x Qwen 2.5 instruct 32B. The choice of the models was based on the availability of drastically different sizes for ready experimentation. Model-strength heterogeneity flipped the Utilitarian Boost: When the deliberating dyad mixes a strong- and a weak-capacity agent, pairs became more deontological, not more utilitarian, at the end of their discussions (β = 1.40, SE = 0.17, z = 8.28, p < .001). The main effect of strength alone was null (β = 0.20, p =.22) and the Group × Strength interaction was likewise non-significant (β = –0.24, *p =.30), showing that the reversal is driven by heterogeneity itself, not by which agent is “strong” or “weak.” Opinion scores did not differ between potent- and weak-capacity agents, meaning the results are not due to baseline differences in judgments across model strengths (β = 0.20, SE = 0.17, z = 1.22, p = .22).
> Together with our earlier findings, these new results converge on the idea that any salient heterogeneity inside an LLM team dampens, and can even overturn, the otherwise robust group-level shift toward utilitarian choices.
>
> [Q3]  a multi-step self-debate for a solo model: would its answer change?
> [RQ3] In a new post hoc experiment, we examined two model variants by comparing their responses at the start of self-reflection versus the last step. The contrast yielded no “utilitarian boost” (p = .63). These null results are preliminary. To fully characterize the dynamics, we will report model-specific trajectories.

---

> > ### Author Response · Authors · 2025-08-06
> > **follow up**
> >
> > We hope this message finds you well. As the discussion phase is coming to a close, we wanted to kindly follow up in case you had any further questions or comments. We would be happy to clarify any concerns or provide additional details that may assist in your evaluation.

---

### Official Review · Reviewer_CAnQ · 2025-07-04

**Clarity:** 3
**Significance:** 3
**Originality:** 2
**Rating:** 4
**Confidence:** 4

**Summary:**

This paper studies moral reasoning in multi-agent systems comprising of large language models (LLMs). It examines whether LLMs exhibit a utilitarian boost, a tendency to endorse norm-violating actions for the greater good when reasoning collectively in small groups (pairs or triads), as compared to reasoning independently. Drawing on paradigms and measures from moral psychology, the authors test six LLMs across moral dilemma benchmarks and analyze the behavioral outcomes and the underlying mechanisms of their moral shifts.

The paper finds that personal moral dilemmas, where agents must decide to directly harm one individual to maximize the utility for others, all models found moral violations to be more acceptable when part of a group than individually, similar to human experiments. Some models endorsed actions that maximized overall well-being, even if they benefited strangers over familiar individuals. Others became more willing to violate moral norms in groups. However, while human groups show a similar action bias, the mechanism for their utilitarian boost differs from LLMs. The paper also finds that that while the surface behavior of LLM collectives mimics human group reasoning, the underlying drivers differ.

**Questions:**

See weaknesses

**Ethical Concerns:**

["NO or VERY MINOR ethics concerns only"]

**Final Justification:**

The response addresses the few concerns I had. In general, I think this is solid work. Not very novel and deep, but interesting and valuable enough to be published.

**Limitations:**

yes

**Quality:**

3

**Strengths And Weaknesses:**

Strengths:

1. The paper extends work on moral reasoning for single LLM to an agentic setting with multiple LLM agents.
2. The paper studies a variety of moral dilllema dimensions - Personal vs. Impersonal, Action vs. Inaction, Impartial Beneficence, Factual Moral Framework and the CNI framework.
3. The paper also tries to find model-specific mechanisms that are driving these moral shifts. Although this direction can be developed better.
4. The experiments are generally good.

Weaknesses:
1. While the main claims of the paper seem to depend on comparisons of models to human data , a direct human-group baseline under the same conditions
2. Some qualitative analysis of the agent interactions could further help explain how the interaction dynamics lead to moral shifts

---

> ### Author Rebuttal · Authors · 2025-07-31
>
> We appreciate the reviewer’s recognition of our shift from single-model to multi-agent moral reasoning, the breadth of dilemma types included, and our efforts to uncover model-specific mechanisms behind moral shifts. Your positive evaluation of the experimental design and framing is deeply encouraging and helps validate the importance of this work for AI safety.
>
> **[W1]** While the main claims of the paper seem to depend on comparisons of models to human data, a direct human-group baseline under the same conditions is missing.
>
> (Note for the (Senior) Area Chairs, the answer R1 below is also given identically to reviewer 32Uz under R3b.)
>
> **[R1]** Our study is focused on alignment issues within multi-agent LLM systems rather than the simulation of human group behavior in similar tasks. We hence believe that the causes and moderators of the utilitarian boost in multi-agent LLMs remain scientifically and practically important regardless of (dis)similarity to human groups. However, we agree with the reviewer that comparisons with humans are essential for understanding the effect. We will add additional information about prior human experiments to the revised manuscript in a study-by-study manner to allow for more fine-grained comparisons. We have summarized these comparisons below.
>
> (i) Keshmirian et al. (2021): individual ratings cluster a little below the scale midpoint (≈ 3.8 / 7). Across the six LLMs, Solo means sit lower (≈ 2.7 – 3.6), so agents start less utilitarian than people. This baseline gap is most significant for GPT‑4.1 (2.66) and smallest for QwQ (3.56). Human groups and every LLM model show the same qualitative pattern: collective discussion pushes judgments toward the utilitarian pole. Quantitatively, Gemma3 produces a larger jump (+0.85) than the human average. The mean boost across models (≈ +0.42) is very close to the human +0.5‑point uptick.
>
> (ii) Cecru et al. (2019): Individual ratings’ mean is 4.92 utilitarian endorsements. Deliberation lifted that to  5.52 (+0.6 points). LLMs show the same qualitative pattern: models’ mean rating increases when in dyads or triads, but the baseline is lower (2.7‑3.6 on the same scale).
>
> (iii)  Rokosz et al (2025): group discussion boosts utilitarianism by increasing only C (outcome) in CNI. Norm focus and omission bias stay stable. Pooling all six models gave a β = 2.12 for the Group > Solo contrast. Humans show the same direction but a smaller magnitude: in Rokosz et al.’s triads, the mean “norm‑violating, outcome‑maximising” score rose from 2.48 ± 1.31 to 3.48 ± 1.31 (0–6 scale). LLM groups amplify utilitarianism equally for lethal and resource dilemmas, though individual models diverge.
>
> **[W2]** Some qualitative analysis of the agent interactions could further help explain how the interaction dynamics lead to moral shifts
>
> **[R2]** Several dimensions of agent interactions were identified by prior psychological research as potentially important determinants of utilitarian versus deontological judgments, including a focus on harm, concreteness, and conversational receptiveness. Detailed results from the analysis of these features in LLM interactions can be found in Section F of the Supplementary Materials, titled Reliability Checks: Argument Analysis. However, we were inspired by the reviewer's comment to look at an additional dimension identified in past research: Emotionality. This post-hoc evaluation provided one of the most promising explanations for the Utilitarian Boost, which we discuss in more detail below.
>
> Psychology and neuroscience have identified emotions as key to the processing of personal versus non-personal moral dilemmas (Greene et al, 2014). We therefore tagged each argument using a RoBERTa-based (Hartmann, 2022) classifier for Ekman’s six emotions plus neutral label. We then correlated those tags with the models’ utilitarian shifts from the Solo to the Group condition. Examining the dominant emotion by experimental condition and dilemma effect revealed an interesting pattern: in the Solo condition, the top emotion label was ‘disgust’, particularly in personal dilemmas where Solo judgments were less utilitarian. In contrast, in the Group condition, dilemmas with no or negative utilitarian shift were predominantly labeled ‘neutral’, while those with firm utilitarian shifts were labeled ‘fear’. Tabulating the emotions by LLM, Gemma3 emerges as having the most ‘fear’ label, while showing the most substantial utilitarian boost (Table 1). This finding aligns with human data: both fear and disgust significantly predict utilitarian biases in personal moral dilemmas in humans. For instance, individuals with poorer disgust decoding are more likely to make utilitarian choices in Personal dilemmas (Martina Carmona-Perera et al, 2014). Stress significantly increased utilitarian decisions when participants faced personal moral dilemmas (Farid Yousef, 2012).
>
> These results highlight emotion as a plausible lever for influencing LLM moral reasoning toward utilitarian outcomes, while proposing a potential mechanism at play. Experimentally attenuating disgust or amplifying fear and anger within model outputs, or human persuasive texts, could plausibly shift moral reasoning along the utilitarian-deontic spectrum. This offers a concrete target for future intervention studies.  We designed prompts that instructed models to adopt a deontological, utilitarian, or neutral moral orientation, then placed them in homogeneous (DD) or mixed pairs (UD, DU). Since we were interested in a Utilitarian Boost and the UU condition could not demonstrate such a result due to expected ceiling effects, we excluded utilitarian‑utilitarian (UU) pairs. We compared the first round of discussion with the group reflection phase response.
>
> Additional novel analyses will be added to the revised manuscript to further explore the interaction dynamics, showing that utilitarian agents are more prone to conformity than deontological ones (D-U= -0.467± 0.0957, p <.0001). This mirrors recent findings in Solo humans (Marton-Alper et al. 2022; Bostyn & Roets, 2017). Pairs instructed to reason deontologically (DD) gave higher ratings after discussions of direct harm, demonstrating the robustness of the Utilitarian Boost (Pair - Round1 = +0.377, p = .010).
>
> We found group composition as moderators: mixed UD/DU pairs became less utilitarian (Pair-Round1 = −0.323, p < .0001), indicating that initial moral framework diversity counters the Utilitarian Boost.  Additional analyses showed that utilitarian agents are more prone to conformity than deontological ones (D-U = -0.467, p < .0001). This, interestingly, mirrors human data in moral conformity: individuals with utilitarian leanings tend to conform both more often and more strongly than their deontological counterparts (Marton-Alper et al. 2022). People tend to conform more readily to deontological than to utilitarian opinions, revealing an asymmetry in moral conformity (Bostyn & Roets, 2017).
>
> Thus, collective moral outcomes of LLM groups are sensitive to the distribution of initial value positions, and a single strategically placed dissenting voice can dampen runaway utilitarianism. This provides practical levers for value‑alignment interventions.

---

> > ### Author Response · Authors · 2025-08-06
> > **Follow up**
> >
> > We hope this message finds you well. As the discussion phase is coming to a close, we wanted to kindly follow up in case you had any further questions or comments. We would be happy to clarify any concerns or provide additional details that may assist in your evaluation.

---

> ### Comment · Reviewer_CAnQ · 2025-08-06
> **Thanks**
>
> Thanks for your comprehensive response! It answers my questions well. I maintain my assessment that this paper is above the acceptance threshold for me.

---

### Note · Authors · 2025-08-14

We thank the AC and the reviewers for their time and effort.

LLM collectives are increasingly tasked with complex decision-making across high-stakes domains. We show that these systems are not neutral aggregators: they produce systematic shifts in moral judgments, reflected in a Utilitarian Boost that emerges across models and persists across robustness checks.

The Utilitarian Boost indicates that LLMs collaborating on moral dilemmas are more willing to violate norms (e.g., killing a human to save many) than when reasoning in solo. Based on the reviews, we ran targeted follow-up experiments reported in the rebuttal and discussion phase. Across models, group sizes, and role compositions, we find a reliable utilitarian boost for groups vs. solo in all six LLMs, with predictable, identifiable moderators (model diversity, role instructions and argument content). These boundary conditions are practical levers: they allow the boost to be amplified, attenuated, or prevented in multi-agent systems as needed.

The results have direct relevance to the LLM and broader AI community:
1. Evaluation validity: Benchmarks involving multiple interacting agents can yield systematically different outcomes from solo benchmarks. Without accounting for this shift, cross-system comparisons and leaderboard rankings may misrepresent capabilities.
2. Systems design: Identifying when the boost is amplified or mitigated, makes it a controllable design parameter in multi-agent systems. This enables deliberate amplification or suppression in low/high-stakes contexts to avoid unintended value drift.
3. Safety and alignment: Social dynamics among models can systematically alter outputs, indicating that alignment efforts must account not only for single-agent reasoning but also for emergent group behaviors.
4. Robustness and reproducibility: The boost persists across architectures and repetitions, providing an example of a reproducible, model-agnostic social effect in LLMs.
5. Broader AI governance: Understanding predictable, design-sensitive social interaction effects is key for setting policy around multi-agent LLM systems deployments in domains like law, policy advising, and autonomous decision-making.

By mapping a social-interaction effect in LLMs, quantifying it across conditions, and offering tested interventions that can amplify or mitigate it, we provide an empirical phenomenon of interest *and* practical guidance for building safer, more predictable multi-agent LLM systems.

---

### Decision · Program_Chairs · 2025-09-17

**Decision:**

Accept (poster)

**Comment:**

The paper studies moral judgment in multi-agent LLM groups and documents a consistent "utilitarian boost" in group decisions relative to solo models, together with analyses that probe why the boost appears and when it attenuates. Three reviewers (CAnQ, 32Uz, 2YLG) are positive (borderline accept), one reviewer (iiw5) remains negative (borderline reject) due to limited mechanistic depth and application framing.

The paper offers a well-executed and timely empirical contribution on collective moral reasoning in LLM groups. With the requested clarifications and explicit documentation of limitations in the camera-ready, the work is suitable for publication and should be valuable to the NeurIPS community.

The rebuttal adds follow-ups add substantive evidence: emotion/CNI-based mechanisms, robustness/boundary conditions (heterogeneous groups, number of rounds, group size (GPT-4.1 tetrads)), a human rater validation of model judgments, and so on.
The rebuttal substantively addresses CAnQ and 32Uz, partially addresses 2YLG, and narrows (though does not eliminate) iiw5’s concerns.
Considering the balance of reviews and the strengthened rebuttal, I recommend Weak Accept this paper.